# Skew-Explore: Learn faster in continuous spaces with sparse rewards

## Abstract

In many reinforcement learning settings, rewards which are extrinsically available to the learning agent are too sparse to train a suitable policy. Beside reward shaping which requires human expertise, utilizing better exploration strategies helps to circumvent the problem of policy training with sparse rewards. In this work, we introduce an exploration approach based on maximizing the entropy of the visited states while learning a goal-conditioned policy. The main contribution of this work is to introduce a novel reward function which combined with a goal proposing scheme, increases the entropy of the visited states faster compared to the prior work. This improves the exploration capability of the agent, and therefore enhances the agent's chance to solve sparse reward problems more efficiently. Our empirical studies demonstrate the superiority of the proposed method to solve different sparse reward problems in comparison to the prior work.

## 1 Introduction

Reinforcement Learning (RL) is based on performing exploratory actions in a trial-and-error manner and reinforcing those actions that result in superior reward outcomes. Exploration plays an important role in solving a given sequential decision-making problem. A RL agent cannot improve its behaviour without receiving rewards exceeding the expectation of the agent, and this happens only as the consequence of properly exploring the environment.

In this paper, we propose a method to train a policy which efficiently explores a continuous state space. Our method is particularly well-suited to solve sequential decision-making tasks with sparse terminal rewards, i.e., rewards received at the end of a successful interaction with the environment. We propose to directly maximize the entropy of the history states by exploiting the mutual information between the history states and a number of reference states. To achieve this, we introduce a novel reward function which, given the references, shapes the distribution of the history states. This reward function, combined with goal proposing learning frameworks, maximizes the entropy of the history states. We demonstrate that this way of directly maximizing the state entropy, compared to indirectly maximizing the mutual information (Warde-Farley et al., 2018; Pong et al., 2019) improves the exploration of the state space as well as the convergence speed at solving tasks with sparse terminal rewards.

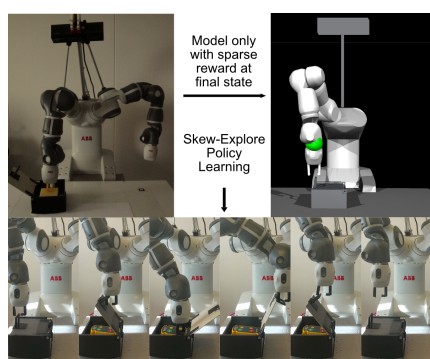

Figure 1: A sparse-reward task trained by a YuMi robot in simulation and deployed to the real hardware. The task consists of (1) opening a door, (2) pressing a button, and (3) closing the door. The reward is given only at the end of a successful trial.

Maximizing the mutual information between the visited states and the goal states, $\mathcal{I}(\mathbf{S}; \mathbf{G})$, results in a natural exploration of the environment while learning to reach to different goal states (Warde-Farley et al., 2018; Pong et al., 2019). The mutual information can be written as $\mathcal{I}(\mathbf{S}; \mathbf{G}) = h(\mathbf{G}) - h(\mathbf{G}|\mathbf{S})$; therefore maximizing the mutual information is equivalent to maximizing the en-

tropy of the goal state while reducing the conditional entropy (conditioned on the goal state). The first term, encourages the agent to choose its own goal states as diverse as possible, therefore improving the exploration, and the second term forces the agent to reach the different goals it has specified for itself, i.e., training a goal-conditioned policy, $\pi(.|s, g)$. Instead of maximizing the mutual information, we propose to maximize the entropy of the visited states directly, i.e., maximizing $h(\mathbf{S}) = h(\mathbf{Z}) + h(\mathbf{S}|\mathbf{Z}) - h(\mathbf{Z}|\mathbf{S})$, where $\mathbf{Z}$ is a random variable that represents the reference points of promising areas for exploration. Therefore, in our formulation, we have an extra term, $h(\mathbf{S}|\mathbf{Z})$, which encourages maximizing the entropy of the state conditioned on the reference points. This extra term, implemented by the proposed reward function, helps the agent to explore better at the vicinity of the references. We call our method **Skew-Explore**, since similar to Skew-Fit introduced by Pong et al. (2019), it skews the distribution of the references toward the less visited states, but instead of directly reaching the goals, it explores the surrounding areas of them.

We experimentally demonstrate that the new reward function enables an agent to explore the state space more efficiently in terms of covering larger areas in less time compared to the earlier methods. Furthermore, we demonstrate that our RL agent is capable of solving long-term sequential decision-making problems with sparse rewards faster. We apply the method to three simulated tasks, including a problem to find a trajectory of a YuMi robot end-effector, to open a door of a box, pressing a button inside the box and closing the door. In this case, the sparse reward is given only when the button is pressed and the door is closed, i.e., at the end of about one minute of continuous interaction with the environment. To validate appropriateness of the trajectory found in simulation, we deployed it on a real YuMi robot, as shown in Figure 1. The main contributions of this paper can be summarized as (1) introducing a novel reward function which increases the entropy of the history states much faster compared to the prior work, and (2) experimentally demonstrating the superiority of the proposed algorithm to solve three different sparse reward sequential decision-making problems.

## 2 RELATED WORK

Prior works have studied different algorithms for addressing the exploration problem. In this section, we summarize related works in the domain where rewards from the environment are sparse or absent.

**Intrinsic Reward:** One way to encourage exploration is to define an intrinsically-motivated reward, including methods that assimilate the definition of curiosity in psychology (Oudeyer et al., 2007; Pathak et al., 2017). These methods have found success in domains like video games (Ostrovski et al., 2017; Burda et al., 2018). In these approaches, the "novelty", "curiosity" or "surprise" of a state is computed as an intrinsic reward using mechanisms such as state-visiting count and prediction error (Schmidhuber, 1991; Stadie et al., 2015; Achiam & Sastry, 2017; Pathak et al., 2017). By considering this information, the agent is encouraged to search for areas that are less visited or have complex dynamics. However, as pointed out by Ecoffet et al. (2019), an agent driven by intrinsic reward may suffer from the problem of detaching from the frontiers of high intrinsic reward area. Due to catastrophic forgetting, it may not be able to go back to previous areas that have not yet been fully explored (Kirkpatrick et al., 2017; Ellefsen et al., 2015). Our method is able to keep tracking the novelty frontier and train policy to explore different areas in the frontier.

**Diverse Skill/Option Discovery:** Methods that aim to learn a set of behaviours which are distinct from each other, allow the agent to interact with the environment without rewards for a particular task. Gregor et al. (2016) introduced an option discovery technique based on maximizing the mutual information between the options and the final states of the trajectories. Eysenbach et al. (2018); Florensa et al. (2017a); Savinov et al. (2018) proposed to learn a fixed set of skills by maximizing the mutual information through an internal objective computed using a discriminator. Achiam et al. (2018) extended the prior works by considering the whole trajectories and introduced a curriculum learning approach that gradually increases the number of skills to be learned. In these works, the exploration is encouraged implicitly through learning diverse skills. However, it is difficult to control the direction of exploration. In our method, we maintain a proposing module which tracks the global information of the states we have visited so far, and keep proposing reference points that guide the agent to the more promising areas for exploration.

**Self-Goal Proposing:** Self-goal proposing methods are often combined with a goal-conditioned policy (Kaelbling, 1993; Andrychowicz et al., 2017), where a goal (or task) generation model is

trained jointly with a goal reaching policy. The agent receives rewards in terms of completing the internal tasks which makes it possible to explore the state space without any supervision from the environment. Sukhbaatar et al. (2017) described a scheme with two agents. The first one proposes tasks by performing a sequence of actions and the other repeats the actions in reverse order. Held et al. (2018) introduced a method that automatically label and propose goals at the appropriate level of difficulty using adversarial training. Similar works are proposed by Colas et al. (2018); Veeriah et al. (2018); Florensa et al. (2017b), where goals are selected based on the learning progress. Warde-Farley et al. (2018) trained a goal-conditioned policy by maximizing the mutual information between the goal states and the achieved states. The goals are selected from the agent's recent experience with strategies. Later, Pong et al. (2019) applied a similar idea of using mutual information. They maximize the entropy of a goal sampling distribution. The focus of these methods is on learning a policy that can reach diverse goals. Although gradually increasing the scale of the goal proposing network, the agent may eventually cover the entire state space, exploration itself is not efficient. In our work, we adopt the same idea of maximizing the entropy of the goal sampling distribution by Pong et al. (2019). However, instead of using the goal-conditioned policy, we introduce a reference point-conditioned policy which greatly increases the efficiency of exploration.

## 3 SKEW-EXPLORE: SEARCHING FOR THE SPARSE REWARD

We discuss the policy learning problem in continuous state and action spaces, which we model as an infinite-horizon Markov decision process (MDP). The MDP is fully characterized by a tuple $(\mathcal{S}, \mathcal{A}, p_{\mathbf{a}}(\mathbf{s}, \mathbf{s}'), R'_{\mathbf{a}}(\mathbf{s}, \mathbf{s}'))$, where $\mathcal{S}$, the state space, and $\mathcal{A}$, the action space, are subsets of $\mathbb{R}^n$, the unknown transition probability $p : \mathcal{S} \times \mathcal{A} \times \mathcal{S} \to [0, \inf)$ indicates the probability density function of the next state $\mathbf{s}'$ given the current state $\mathbf{s} \in \mathcal{S}$ and the action $\mathbf{a} \in \mathcal{A}$. For each transition, the space associated environment $\mathcal{E}$ emits an extrinsic reward according to function $R' : \mathcal{S} \times \mathcal{A} \to \mathbb{R}$. The objective of the agent is to maximize the discounted return, i.e. *return* $R = \sum_{t_s=0}^{\infty} \gamma^{t_s} r_{t_s}$, where $\gamma$ is a discounted factor and $r_{t_s}$ is the reward received at each step $t_s$. In this study, we consider an agent interacting in an environment $\mathcal{E}$ with *sparse reward*. The *sparse reward* $r$ is modelled as a truncated Gaussian function with a narrow range. From previous interactions, the agent holds an interaction set $\mathcal{I}_t$, in which transaction triples $(\mathbf{s}_j, \mathbf{a}_j, \mathbf{s}_{j+1}), \forall j \in \{1, \cdots, T-1\}$ are contained. We also extract the states $s_j$ from $\mathcal{I}_t$ to form a **history state set** $\mathcal{S}_t$, which contains all visited states by the agent until iteration $t$. The objective of our method is to find an arbitrary external goal in a continuous state space and converge to a policy that maximizes the $R$ as fast as possible. This involves two processes 1) Find the external reward through efficient exploration. 2) Converge to a policy that maximizes $R$ once the external reward is found.

We can use the entropy of the history state set as a neutral objective to encourage exploration, since an agent that maximizes this objective should have visited all valid states uniformly. To describe it mathematically, we define a random variable $\mathbf{S}$ to represent the history states that the agent has visited. The distribution of $\mathbf{S}$ is estimated from the history state set $\mathcal{S}_t$. Our goal is to encourage exploration by maximizing the entropy $h(\mathbf{S})$ of the history states. However, using the entropy as the intrinsic reward directly may suffer from problems similar to other intrinsic motivated methods (Schmidhuber, 1991; Stadie et al., 2015; Achiam & Sastry, 2017; Pathak et al., 2017). As the reward of the same state is changing, the agent has the risk of detaching from the frontiers of high intrinsic reward area.

We introduce a concept called *novelty frontier reference point*, which can be sampled from a distribution that represents the *novelty frontier* (Ecoffet et al., 2019). The novelty frontier defined in our work represents the areas near the states with lower density in distribution $p(\mathbf{s})$. The frontier reference points are sampled after the distribution of the novelty frontier is updated. We define a $\mathbf{Z}$ to represent all the history frontier reference points with probability density $p(\mathbf{z})$ estimated from a set $\mathcal{Z}_t$ that contains all novelty frontier reference points until iteration $t$.

The conditional probability $p(\mathbf{s}|\mathbf{z})$ defines the behaviour of the agent with respect to each reference point. In this work, we model this behaviour using a state distribution function $K_z(\mathbf{s} - \mathbf{z})$ parameterized by the displacement between the state and the reference point. The function $K_z$ needs to be chosen carefully as it should satisfy our expectation of the policy behaviour and also, provides an informative reward signal to train the policy. Mathematically, we can rewrite $p(\mathbf{s})$ as $p(\mathbf{s}) = \int f(\mathbf{s}|\mathbf{z})p(\mathbf{z})d\mathbf{z} = \int K_z(\mathbf{s} - \mathbf{z})p(\mathbf{z})dz$. Generally, $K_z(\cdot)$ can be different for different $\mathbf{z}$.

However, to reduce the complexity of learning, we constrain $K_z(\cdot)$ to be consistent for any $\mathbf{z}$, meaning $K_z(\mathbf{s}-\mathbf{z}) = K(\mathbf{s}-\mathbf{z})$. The definition of $K(\cdot)$ satisfies the definition of a kernel function. Using $K(\mathbf{s}-\mathbf{z})$, $p(\mathbf{s})$ can then be further represented as

$$p(\mathbf{s}) = \int K(\mathbf{s} - \mathbf{z})p(\mathbf{z})d\mathbf{z}$$
$$= (K * p)(\mathbf{s}).$$

By considering the law of convolution of probability distributions, we obtain $\mathbf{S} = \mathbf{Z} + \mathbf{N}$, where $\mathbf{N}$ is a random variable characterized by a density function $K(\cdot)$. Now with this setup, we are able to to analyze our method's performance using information theory. By considering the entropy's relationship with mutual information $h(\mathbf{S}) = h(\mathbf{S}|\mathbf{Z}) + \mathcal{I}(\mathbf{S};\mathbf{Z})$, we receive the final decomposition of our objective under the novelty frontier reference point-conditioned policy framework

$$h(\mathbf{S}) = h(\mathbf{Z}) + h(\mathbf{S}|\mathbf{Z}) - h(\mathbf{Z}|\mathbf{S}). \tag{1}$$

Eq. 1 indicates that in order to maximize the $h(\mathbf{S})$, we can individually maximize/minimize each term while making other terms fixed. In the following section, we will explain the optimization process in detail.

### 3.1 MAXIMIZING $h(\mathbf{Z})$: OBTAINING AN EXPANDING SET OF NOVELTY FRONTIER REFERENCE POINTS

As introduced above, $h(\mathbf{Z})$ is the entropy estimated from the novelty frontier reference points set $\mathcal{Z}_t$. To increase $h(\mathbf{Z})$, we need to add a new reference point to $\mathcal{Z}_t$ such that, the entropy estimated form $\mathcal{Z}_{t+1}$ is larger than the entropy estimated from $\mathcal{Z}_t$. In our method, the frontier reference points are sampled from the novelty frontier distribution which represents less history areas according to the current history states. Pong et al. (2019) proposed a method to skew the distribution of the history states using importance sampling, such that states with lower density can be proposed more often. In our work, we use a similar way to estimate the novelty frontier distribution. There are three steps in our process. In the first step, we estimate the $p(\mathbf{s})$ from $\mathcal{S}_t$ using a density estimator e.g. Kernel Density Estimation (KDE). In the second step, we sample $Q$ states $\{\mathbf{s}_0, \cdots, \mathbf{s}_Q\}$ from $p(\mathbf{s})$, and compute the normalized weight for each state using Eq. 2

$$w_i = \frac{1}{Y_\alpha}p(\mathbf{s}_i)p(\mathbf{s}_i)^\alpha \qquad \alpha \in [-\inf, 0), Y_\alpha = \sum_{n=1}^{N} p(\mathbf{s} = \mathbf{s}_Q)p(\mathbf{s} = \mathbf{s}_Q)^\alpha, \tag{2}$$

where $Y_\alpha$ is a normalizing constant. The state with lower $p(\mathbf{s})$ has higher weight and vice versa. Finally, we utilize a generative model training scheme $T_g(\cdot, \cdot)$ (e.g. weighted KDE), together with sampled states and weights to get a skewed distribution $p_{skewed}(\mathbf{s}) = T_g(\{\mathbf{s}_0, \cdots, \mathbf{s}_n\}, \{w_0, \cdots, w_n\})$ to represent the novelty frontier distribution.

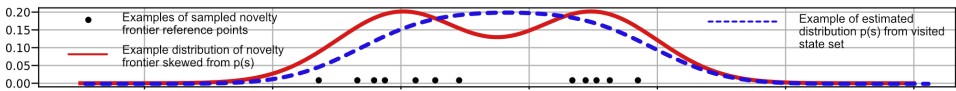

Figure 2: This figure shows a comparison between the state distribution $p(\mathbf{s})$ (dashed blue) and a corresponding novelty frontier distribution skewed from $p(\mathbf{s})$ (red).

If $Q$ is big enough, by choosing a $\alpha$ appropriately, we are able to expand our frontiers after each iteration. As a consequence, the distribution estimated from $\mathcal{Z}_t$ will become more and more uniform and its range will become larger and larger, just like annual ring of the tree. The entropy of a continuous uniform function $U(p,q)$ is $\ln(p-q)$ and if the distribution has a larger range, the entropy is larger as well. Fig 2 illustrates the estimated frontier distribution skewed from $p(\mathbf{s})$.

### 3.2 MAXIMIZING $h(\mathbf{S}|\mathbf{Z}) - h(\mathbf{Z}|\mathbf{S})$: INCREASING THE EXPLORATION RANGE AROUND REFERENCE POINTS

The conditional entropy of $h(\mathbf{S}|\mathbf{Z})$ and $h(\mathbf{Z}|\mathbf{S})$ are highly correlated, maximizing/minimizing them individually are difficult. Therefore, in this section, we consider to maximize $h(\mathbf{S}|\mathbf{Z})$ - $h(\mathbf{Z}|\mathbf{S})$

as a whole. Using the relation $\mathbf{S} = \mathbf{Z} + \mathbf{N}$, we rewrite the expression as $h(\mathbf{S}|\mathbf{Z}) - h(\mathbf{Z}|\mathbf{S}) = h(\mathbf{Z} + \mathbf{N}|\mathbf{Z}) - h(\mathbf{Z}|\mathbf{Z} + \mathbf{N})$, which can be further simplified (see Appendix D) as

$$h(\mathbf{Z}|\mathbf{S}) - h(\mathbf{S}|\mathbf{Z}) \geq h(\mathbf{N}) - h(\mathbf{Z}).$$

This implies that there is a lower bound for the expression $h(\mathbf{S}|\mathbf{Z}) - h(\mathbf{Z}|\mathbf{S})$. For a fixed $h(\mathbf{Z})$, we can maximize the lower bound $h(\mathbf{N}) - h(\mathbf{Z})$ by increasing $h(\mathbf{N})$. $h(\mathbf{N})$ is related to the shape and variance of the exploration distribution near the reference point. In our method, we model $\mathbf{N}$ as a Gaussian distribution with zero mean. In an ideal case, we would like to have as large variance as possible. However, increasing the variance also results in learning difficulty, as we need a longer trajectory to evaluate the performance and more samples to update the network. Therefore, we use the variance to control the trade-off between exploration efficiency and learning efficiency.

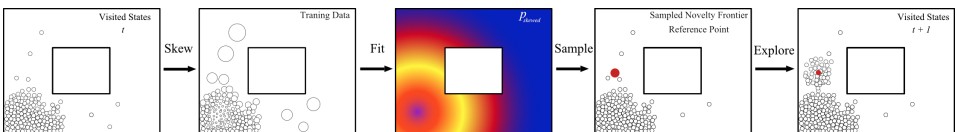

Figure 3: Our method, Skew-Explore, aims to obtain uniform state visitation distribution estimated from the history state set. We start by sampling from our history state set, and weighting the states such that less-visited states are assigned with higher weights. We then train a skewed distribution $p_{skewed}(\mathbf{s})$ using the weighted samples as the novelty frontier distribution. Next we sample reference points on the novelty frontier distribution and run our policy to explore around the points.

### 3.3 Designing the reward function

Our algorithm requires the policy to move around a given reference point, and the distribution of the states in the trajectory should follow a Gaussian distribution centered at the reference point. In this section, we introduce an intrinsic reward function to train such policy by minimizing the the Kullback-Leibler (KL) divergence between the trajectory distribution and the desired Gaussian distribution. For each given reference point $\mathbf{z}_i$, we collect a trajectory $\tau_i$ by running the policy with the given reference point, indicated as $\pi(\mathbf{z}_i)$, for $M$ steps. Then, we estimate the probability density of each state $\mathbf{s}$ in $\tau_i$, referred to as $p_{\tau_i}(\mathbf{s})$, using a density estimator. Finally, we check the probability density of $\mathbf{s}$ in the Gaussian distribution centered at $\mathbf{z}_i$, referred to as $p_{\mathbf{z}_i}(\mathbf{s})$. The KL-divergence between the trajectory distribution $p_{\tau_i}$ and the desired distribution $p_{\mathbf{z}_i}$ is formulated as follows

$$\mathcal{D}_{KL}(p_{\tau_i}(\cdot) \mid p_{z_i}(\cdot)) = \mathbb{E}_{\mathbf{z}_i \sim (\mathcal{Z}_t), \tau_i \sim \pi(\mathbf{z}_i), \mathbf{s} \sim \tau_i} p_{\tau_i}(\mathbf{s}) \log \frac{p_{\tau_i}(\mathbf{s})}{p_{\mathbf{z}_i}(\mathbf{s})}. \tag{3}$$

To minimize the KL-divergence, the intrinsic reward of $\mathbf{s}$ with respect to $\mathbf{z}_i$ is computed as

$$r_{int}(\mathbf{s}, \mathbf{z}_i) = \log(p_{\mathbf{z}_i}(\mathbf{s})) - \log(p_{\tau_i}(\mathbf{s}))). \tag{4}$$

The intrinsic reward function measures the difference between the desired density of $\mathbf{s}$ in the trajectory and the actual density achieved. The reward is positive when the actual density is smaller than the desired one, when states in the trajectory are too far from the reference point, and the reward is negative when the actual density is larger than the desired one, when the agent stays too long at the reference point. An extrinsic reward $r_{ext}(\mathbf{s})$ is provided by the environment and the total reward of a time step is defined as the weighted sum of the intrinsic and the extrinsic reward. The extrinsic reward should be much greater than the intrinsic reward. The reward of each time step $r(\mathbf{s}, \mathbf{z}_i)$ is defined as

$$r(\mathbf{s}, \mathbf{z}_i) = w_{int} \cdot r_{int}(\mathbf{s}, \mathbf{z}_i) + w_{ext} \cdot r_{ext}(\mathbf{s}), \tag{5}$$

where, $w_{int}$ and $w_{ext}$ are respective weights for internal and external rewards. The performance of the policy is closely related to the set $\mathcal{Z}_t$, as it records the reference points we used to train the policy until iteration $t$. As described in section 3.1, while we increase the entropy $h(\mathbf{Z})$ by proposing new reference points form the novelty frontier to train the policy, the policy gradually gain skills to explore different areas. When a state with a large extrinsic reward is discovered, the policy eventually ignores all given reference points and converge to reach the state with the extrinsic reward. Algorithm 1 shows the whole Skew-Explore algorithm using pseudo code and our implementation of the algorithm is available online [1].

---

[1] https://anonymous.4open.science/r/b4596073-4cbc-4ac6-b85b-e9a786909058/

---

**Algorithm 1** Skew-Explore

---

1: **procedure** SKEW-EXPLORE
2:  History state set $\mathcal{S}_0 = \{\}$
3:  History novelty frontier reference points set $\mathcal{Z}_0 = \{\}$
4:  Randomly sample $L$ novelty frontier points $z_i$
5:  $\mathcal{Z}_0 = \mathcal{Z}_0 \cup \{z_1, ..., z_L\}$.
6:  **for** $t = 1, 2, 3...$ **do**
7:    Collect a set of states $\mathbf{s}_q$ by running policy giving different frontier reference point $\mathbf{z}_i$.
8:    Compute reward for each state using Eq. 5.
9:    Update history state set $\mathcal{S}_t = \mathcal{S}_{t-1} \cup \{\mathbf{s}_1, ..., \mathbf{s}_Q\}$ and estimate $p(\mathbf{s})$ from $\mathcal{S}_t$
10:    Estimate the novelty frontier distribution $p_{skewed}(\mathbf{s})$ by skewing $p(\mathbf{s})$.
11:    Sample $L$ new novelty frontier points $\mathbf{z}_l \sim p_{skewed}(s)$.
12:    Update history novelty frontier reference points set $\mathcal{Z}_t = \mathcal{Z}_{t-1} \cup \{\mathbf{z}_1, ..., \mathbf{z}_L\}$.
13:    Update policy according to the rewards

---

### 3.4 SCALING TO HIGHER DIMENSIONAL STATES

## 4 EXPERIMENT

In this section, we evaluate our algorithm from three perspectives. 1) How efficient is our algorithm in terms of exploring the entire state space, and how different choice of variance affects the efficiency? 2) Is our algorithm able to converge to a stable solution for tasks with sparse reward? 3) Is our algorithm able to solve a complicated sparse reward task with long horizon? The implementation details of the experiments can be found in Appendix E. Two metrics are considered to evaluate the performance. They are the state distribution entropy $h(\mathbf{S})$ and the *coverage*, which are estimated from history state set $\mathcal{S}_t$. We describe how we estimate the two metrics and their difference in Appendix A and B. A short video regarding the experiments can be found online [2].

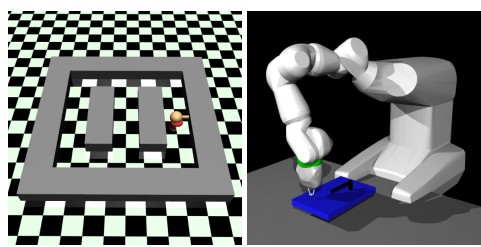

Figure 4: The PointMaze environment and the DoorOpen environment.

### 4.1 POWER OF EXPLORATION

In the first experiment, we evaluate our algorithm in term of the efficiency of exploring the state space. We test our algorithm in two simulated environments, the PointMaze and the DoorOpen environments (Fig. 4). In the PointMaze environment, a point agent is controlled to move inside a maze with narrow passages. In the DoorOpen environment, a YuMi robot can open a door by grabbing the door handle. The PointMaze environment was previously used by Florensa et al. (2017a); Eysenbach et al. (2018); Pong et al. (2019), whilst environments similar to the DoorOpen environment were used by Kalakrishnan et al. (2011); Chebotar et al. (2017); Pong et al. (2019). The objective of the tasks is to explore the entire state space in a minimum amount of time. In order to evaluate the performance, we measure the efficiency as the overall coverage and the entropy of the density estimated from all history states. We compare our algorithm with two baseline algorithms: the random network distillation (RND) proposed by Burda et al. (2018) which is an approach using prediction error as the intrinsic reward, and Skew-Fit proposed by Pong et al. (2019) which combines a goal proposing network with a goal-conditioned policy.

We consider two configurations. The first one is the proximal policy optimization (PPO) Schulman et al. (2017) together with a Long Short-Term Memory network (LSTM) Hochreiter & Schmidhuber (1997). The second configuration is soft actor-critic (SAC) Haarnoja et al. (2018) and hindsight experience replay (HER) Andrychowicz et al. (2017). We note here that RND is only tested with PPO and LSTM as it was not designed for off-policy methods. For each configuration, we run

---

[2]https://www.dropbox.com/s/xxw7ug3lnud3h0j/video_submission.mp4

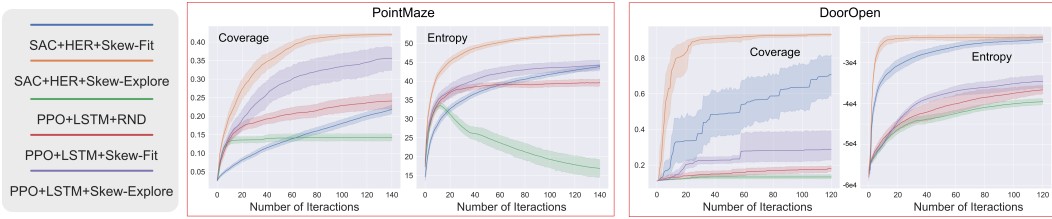

Figure 5: Results of how coverage and entropy changes over iterations in PointMaze (left) and the DoorOpen (right) environments.

12 times and compare the mean and variance. Fig. 5 shows the results for all 5 configurations. We can see that our method, SAC+HER+Skew-Explore, makes both coverage and entropy increase faster than the other methods. It also increases with relatively small variances. Fig 6 illustrates

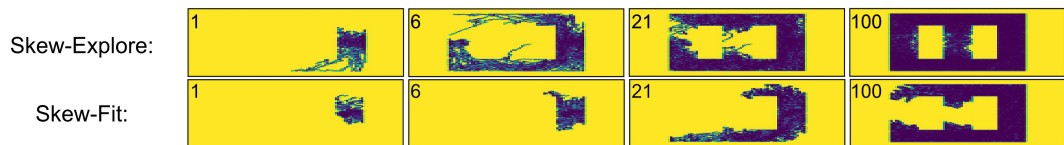

Figure 6: Results of how coverage changes over iterations in the PointMaze environment.

how coverage changes for both our method Skew-Explore and Skew-Fit. In this figure, we see how our method is able to cover the state space (area in this case) faster than Skew-Fit. To further analyze how different choices of the variance of $\mathbf{N}$ affects the exploration efficiency, several values of variances $0^2$, $1^2$, $2^2$, $3^2$, $4^2$ and $6^2$ are tested in the PointMaze environment. After 80 iterations, estimated entropy $40.2\pm1.2$, $49.3\pm0.7$, $51.4\pm0.8$, $51.4\pm0.7$, $51.4\pm0.7$ and $48.9\pm3.3$ are received. We observe that while the variance increases, the performance first increases and then decreases.

## 4.2 POWER OF SOLVING A SINGLE SPARSE REWARD TASK

The power of exploration is an important aspect, but we also want our algorithm to converge to a stable policy that maximizes the extrinsic reward for different sparse reward tasks. To this end, we use the same environments (Fig. 4) as in the previous experiment. In each environment, we select five uniformly distributed target points from the area of interest and assign extrinsic rewards when the agent reaches these points. For each target point, we train an individual policy to reach it. Hypothetically, influenced by the extrinsic reward, the agent eventually ignores the internal

| Environment \ Targets | | Target Point 1 | Target Point 2 | Target Point 3 | Target Point 4 | Target Point 5 |
|---|---|---|---|---|---|---|
| PointMaze | (90±3%) | 349 | 309 | 1975 | 2233 | 2388 |
| DoorOpen | (90±3%) | 415 | 353 | 501 | 321 | 503 |

Table 1: This shows the number of trajectories needed for the algorithm to converge to five uniformly sampled target states in each environment. A successful convergence is measured as 90% of the states receive the extrinsic reward with a standard deviation of less then 3%. The two other methods Skew-Fit and RND cannot solve the problem below upper limit 3000 trajectories.

goals generated by the goal proposing module and reaches the target points consistently. In order to evaluate the performance of our algorithm, we measure how reliably the agent is able to reach each target point. To this end, we collect the final 10 states from the 10 most recent trajectories and define criteria for convergence as the percentage of receiving the extrinsic rewards. If more than 90% of the states receive the extrinsic reward with a standard deviation of less then 3%, we say the agent solved the task successfully. In this experiment, we use the configuration SAC+HER+Skew-Explore which achieves the best performance in the previous experiment. Table 1 shows how many trajectories the algorithm needs to reach the criteria of convergence. This experiment thus shows that our algorithm is able to solve a sparse reward task, by obtaining a policy with a limited number of trajectories. Additional results can be found in Appendix F.

### 4.3    TASK WITH A LONG HORIZON AND REAL WORLD DEMONSTRATION

In the third experiment, we evaluate the ability of our algorithm in terms of solving a sparse reward task with a long horizon and test the performance of the converged policy using a real world YuMi robot. We increase the complexity of the environment by adding a box and a button to the DoorOpen environment used in the previous two experiments. We design a task called OpenPressClose which needs a long sequence of procedures to be solved. The sequence includes 1) open the box, 2) press the button inside the box and 3) close the box. The extrinsic sparse reward is only given to the agent after all procedures in the sequence are done. This task is exceptionally challenging as each intermediate procedure requires a set of continuous actions in correct order to be achieved and no intermediate reward is provided to guide the search. Therefore this task requires the power of efficient exploration to discover the state that provides an extrinsic reward. If the algorithm fails to explore efficiently, the rewarding state would never be found and no policy will be learned. The results show that the algorithm is able to discover the extrinsic reward and converge to a stable solution. Fig 7a shows the change of average extrinsic reward per step over iterations and Fig 7b shows the converged policy in sequential order. The resulting policy is deployed to a real world YuMi robot as shown in Fig 1. A demonstration of the real robot solving the task can be found in the video.

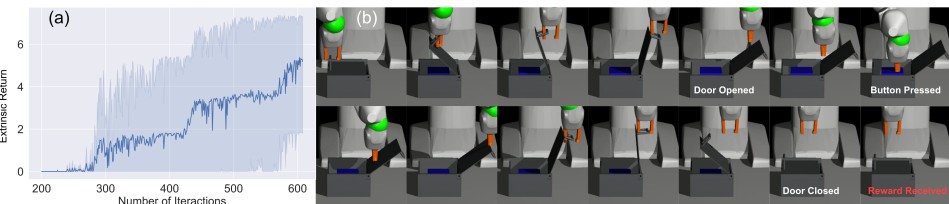

Figure 7: (a) Relation between average extrinsic reward per iterations. (b) From up to down, left to right, this figure shows a sequence of the converged policy for the OpenPressClose task, which is a long horizon task with a sparse reward given only at the end.

## 5    CONCLUSION

In this work, we propose an algorithm named Skew-Explore, a general framework for continuous state exploration. Inspired by Skew-Fit Pong et al. (2019), the main idea of Skew-Explore is to encourage exploration around the novelty frontier reference points proposed by a latent variable proposing network. The algorithm is able to track the global information of entropy of density distribution estimated by the states stored in a history state set, which helps to maximize a corresponded metrics, namely entropy and coverage. Two experiments are conducted to test the power of Skew-Explore on the exploration problem and the single sparse reward problem. In the first experiment, we found that our algorithm Skew-Explore, using SAC and HER together, has the fastest exploration rate. In the second experiment, we found that our algorithm is also able to converge to a stable policy when a single sparse reward is given. As a demonstrator, we used an environment where a robotic manipulator needs to 1) open a door, 2) press a button inside and 3) close the door in a sequence but only with a sparse reward given at the end. We implemented the fully converged policy on a real YuMi robot using policy transfer. Future work will include investigating if we can improve the efficiency of policy convergence by adjusting the proposing network's distribution. Additionally, we will examine whether clustering can increase the efficiency for exploration. Moreover, we will look for a better reward function than KL divergence between Gaussian-based goal distribution and the trajectory distribution.

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

# Appendices

## A   DIFFERENCE BETWEEN COVERAGE AND $h(\mathbf{S})$

Since the reward is given to an arbitrary state in the whole space, two metrics could be considered to evaluate the performance of the agent, namely the state distribution entropy $h(\mathbf{S})$ and the *coverage* $f_c(\mathcal{S}_t)$. The distribution of $\mathbf{S}$ can be estimated using any density estimator trained using $\mathcal{S}_t$.

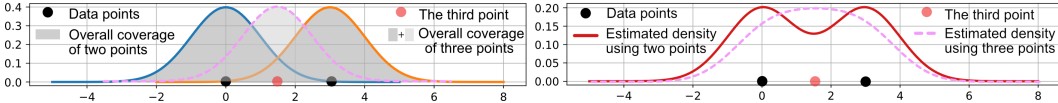

Figure 8: The change of coverage (left) and estimation of probability density (right) when adding a new point, in one-dimensional state space. In the left image, the overall state coverage constructed by two black points is represented as the area of dark grey. When a third point (red dot) is added, the overall state coverage increases and the amount of increase is the area of light grey. In the right image, the red curve shows the density distribution constructed by two black points. When a new point (red dot) is added, the density distribution changes to the dashed curve. The state entropy decreases as the new point is in a region with high density.

The coverage of $\mathcal{S}_t$ is defined as a mapping $f_c : \mathcal{S}^n \to \mathbb{R}$, $K(\mathcal{S}_t) := \int_{\mathcal{S}} \max_i(\phi(\mathbf{s}_i))\,d\mathbf{s}$, where $\phi : [0, \inf] \to \mathbb{R}$ is a radial basis function with a Gaussian kernel centred at $\mathbf{s}_i \in \mathcal{S}_t$, and $n$ equals $|\mathcal{S}_t|$. Intuitively, these two measures describe similar characteristics of the data distribution. However, they also have significant differences. One major difference is that when a new data point is presented in the state space, the coverage can only increase whilst the entropy can decrease if the new data point is given in an often visited area. Figure 8 illustrates the difference between coverage and density estimation when two and three points are given. Coverage gives a better intuition about where the agent should search for the sparse reward.

To find the sparse reward, we maximize the history states entropy to increase the coverage. A proof for one-dimensional state space is given in Appendix C. Proof of high dimensional cases is related to Kepler theorem Hales (2005) and is beyond the scope of this paper. We aim to maximize the entropy of the state density in the $\mathcal{S}_t$, which is $\mathcal{H}(\mathbf{S})$.

## B   ESTIMATING STATE ENTROPY AND COVERAGE

In practice, we uniformly discretize the entire state space along each dimension and use the center of the discretized grids to estimate the entropy and the coverage. We do not check the validity of each grid. Therefore, there are grids that are not reachable from the initial state, and the maximum possible coverage is less than 1. The PointMaze environment is discretized to $50 \cdot 50$ grids. The DoorOpen environment is discretized to $10 \cdot 10 \cdot 10 \cdot 2 \cdot 10$ grids. The entropy and coverage are summed over the discretized states.

## C   PROOF OF RELATION BETWEEN COVERAGE AND $h(\mathbf{S})$

In this section, we prove Lemma C.1.

**Lemma C.1.** Given $N$ movable points $\mathcal{P} = \{p_1, p_2, \cdots, p_N\}$ distributed in $\mathcal{R}$ with boundary $B_l \leq p_i \leq B_h$. Maximizing the coverage estimated from these points using a Gaussian kernel will make the points to be uniformly distributed within the boundary, thus maximizing the entropy of the points' distribution.

Proof. According to the definition of coverage, the area under all Gaussian kernels could be written as :

$$f_c(\mathcal{P}) = \int_{\mathcal{R}} \max_i(\phi(p_i))\,d\mathbf{x} \tag{6}$$

We rewrite the equation as subtraction of all the areas and the overlapping areas. Fig 9 illustrates these concepts. The area from negative infinity to $x$ under Gaussian distribution is defined as.

$$\Phi(x) = \int_{-\infty}^{x} \phi(x)dx = \frac{1}{2}\left(1 + \text{erf}\left(\frac{x}{\sqrt{2}}\right)\right),$$ (7)

$$f_c(\mathcal{P}) = N\Phi(\infty) - \sum_{i=1}^{N-1} 2\Phi(\frac{-t_i}{2})$$ (8)

$$= 1 - \sum_{i=1}^{N-1} \text{erf}\left(\frac{-t_i}{2\sqrt{2}}\right)$$ (9)

$$(10)$$

As maximizing $1 - \sum_{i=1}^{N-1} \text{erf}\left(\frac{-t_i}{2\sqrt{2}}\right)$ is equivalent to minimizing $\sum_{i=1}^{N-1} \text{erf}\left(\frac{-t_i}{2\sqrt{2}}\right)$. We formulate an optimization problem as follows:

$$\underset{t_1, t_2, \cdots, t_i}{\text{minimize}} \quad \sum_{i=1}^{N-1} \text{erf}\left(\frac{-t_i}{2\sqrt{2}}\right)$$

$$\text{subject to} \quad t_i \geq 0, \forall t_i$$

$$\sum_{i=1}^{N-1} t_i \leq B_h - B_l,$$

Solving this problem using the KKT conditions will give us that when $t_i = \frac{B_h - B_l}{N-1}, \forall t_i$, the expression is minimized. This result shows that when the coverage estimated from these points is maximized, the points need to be distributed evenly within the boundary. As a consequence, the entropy of the distribution of these points is also maximized.

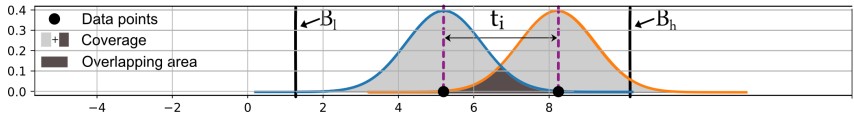

Figure 9: This figure demonstrates the relationship between overall coverage and overlapping area.

## D   PROOF OF $h(\mathbf{S}|\mathbf{Z}) - h(\mathbf{Z}|\mathbf{S}) \geq h(\mathbf{N}) - h(\mathbf{Z})$

In this section, we show our analysis and proof of the expression

$$h(\mathbf{S}|\mathbf{Z}) - h(\mathbf{Z}|\mathbf{S}) \geq h(\mathbf{N}) - h(\mathbf{Z}).$$

Let $\mathbf{Z}$ and $\mathbf{S}$ be jointly distributed continuous random variables, where $\mathbf{S}$ is related to $\mathbf{Z}$ through a conditional PDF $f(s|z)$ defined for all z. The conditional PDF $f(s|z)$ follows a Gaussian distribution centred at different $z$ with standard deviation $\sigma > 0$.

**Lemma D.1.** $\mathbf{S}$ can be represented as the sum of $\mathbf{Z}$ and $\mathbf{N}$, where $\mathbf{N}$ is a r.v. with Gaussian PDF of 0 mean and standard deviation $\sigma$, independent from $\mathbf{Z}$.

$$\mathbf{S} = \mathbf{Z} + \mathbf{N}$$ (11)

**Proof**

Using the law of total probability, the density of $f(\mathbf{s})$ can be written as:

$$
\begin{aligned}
f(\mathbf{s}) &= \int f(\mathbf{s}|\mathbf{z})f(\mathbf{z})d\mathbf{z} \\
&= \int \frac{1}{\sigma\sqrt{2\pi}} e^{-(s-z)^2/2\sigma^2} f(z)dz \\
&= \int K(s-z)f(z)dz \\
&= (K * f)(s),
\end{aligned}
\tag{12}
$$

where $K$ is the density function of $\mathbf{N}$. According to Theorem 7.1 in Grinstead & Snell (2012), when the density function $f(s)$ is the convolution of the density function of $\mathbf{G}$ and $\mathbf{N}$, $\mathbf{S} = \mathbf{G} + \mathbf{N}$ holds.

**Q.E.D.**

**Lemma D.2.** When $\mathbf{S} = \mathbf{Z} + \mathbf{N}$ holds, the following expression

$$
h(\mathbf{S}|\mathbf{Z}) - h(\mathbf{Z}|\mathbf{S}) \geq h(\mathbf{N}) - h(\mathbf{Z})
\tag{13}
$$

holds.

**Proof.**

*Remark.* Here we note that we will use following equations in later proof,

$$
h(\mathbf{Z}) = h(\mathbf{Z}|\mathbf{N}) = h(\mathbf{Z} + \mathbf{N}|\mathbf{N})
\tag{14}
$$
$$
h(\mathbf{N}) = h(\mathbf{N}|\mathbf{Z}) = h(\mathbf{Z} + \mathbf{N}|\mathbf{Z}).
\tag{15}
$$

In Eq. 14 and Eq. 15, $h(\mathbf{Z}) = h(\mathbf{Z}|\mathbf{N})$ and $h(\mathbf{N}) = h(\mathbf{N}|\mathbf{Z})$ are true because $\mathbf{Z}$ and $\mathbf{N}$ are independent. Moreover, $h(\mathbf{Z}|\mathbf{N}) = h(\mathbf{Z} + \mathbf{N}|\mathbf{N})$ and $h(\mathbf{N}|\mathbf{Z}) = h(\mathbf{Z} + \mathbf{N}|\mathbf{Z})$ are true because there is no gain of information by adding what is given already.

We expand the Eq. 13 by replacing the $\mathbf{S}$ with $\mathbf{Z} + \mathbf{N}$ and get

$$
\begin{aligned}
h(\mathbf{S}|\mathbf{Z}) - h(\mathbf{Z}|\mathbf{S}) &= h(\mathbf{Z} + \mathbf{N}|\mathbf{Z}) - h(\mathbf{Z}|\mathbf{Z} + \mathbf{N}) \\
&= h(\mathbf{N}) - h(\mathbf{Z}|\mathbf{Z} + \mathbf{N})
\end{aligned}
\tag{16}
$$

Now we proceed to prove that $h(\mathbf{Z}|\mathbf{Z} + \mathbf{N}) = h(\mathbf{N}|\mathbf{Z} + \mathbf{N})$,

$$
\begin{aligned}
h(\mathbf{Z}|\mathbf{Z} + \mathbf{N}) &= h(\mathbf{G} + \mathbf{N}|\mathbf{Z}) + h(\mathbf{Z}) - h(\mathbf{Z} + \mathbf{N}) \\
&= h(\mathbf{N}) + h(\mathbf{Z}) - h(\mathbf{Z} + \mathbf{N}) \\
&= h(\mathbf{N}) + h(\mathbf{Z} + \mathbf{N}|\mathbf{N}) - h(\mathbf{Z} + \mathbf{N}) \\
&= h(\mathbf{N}, \mathbf{Z} + \mathbf{N}) - h(\mathbf{Z} + \mathbf{N}) \\
&= h(\mathbf{N}|\mathbf{Z} + \mathbf{N}).
\end{aligned}
\tag{17}
$$

We replace $h(\mathbf{Z}|\mathbf{Z} + \mathbf{N})$ with $h(\mathbf{N}|\mathbf{Z} + \mathbf{N})$ in Eq. 16 to have

$$
\begin{aligned}
h(\mathbf{S}|\mathbf{Z}) - h(\mathbf{Z}|\mathbf{S}) &= h(\mathbf{N}) - h(\mathbf{N}|\mathbf{Z} + \mathbf{N}) \\
&= I(\mathbf{Z} + \mathbf{N}; \mathbf{N}) \\
&= I(\mathbf{S}; \mathbf{N}).
\end{aligned}
\tag{18}
$$

According to the property of mutual information, $I(\mathbf{S}; \mathbf{N}) \geq 0$, with equality if and only if $\mathbf{S}$ and $\mathbf{N}$ are independent. As $\mathbf{S} = \mathbf{Z} + \mathbf{N}$, $I(\mathbf{S}; \mathbf{N}) = 0$ is true only when the PDF of $\mathbf{N}$ is a Dirac delta function. In our case, PDF of $\mathbf{N}$ is a Gaussian function and $\sigma > 0$, as a consequence, the expression $h(\mathbf{S}|\mathbf{Z}) - h(\mathbf{Z}|\mathbf{S}) > 0$ holds. Additionally, since $h(\mathbf{Z}|\mathbf{Z} + \mathbf{N})$ is the same as $h(\mathbf{N}|\mathbf{Z} + \mathbf{N})$, by law of conditioning reduces entropy, we has an inequality as follows

$$
\begin{aligned}
h(\mathbf{S}|\mathbf{Z}) - h(\mathbf{Z}|\mathbf{S}) &= h(\mathbf{N}) - h(\mathbf{N}|\mathbf{Z} + \mathbf{N}) \\
h(\mathbf{S}|\mathbf{Z}) - h(\mathbf{Z}|\mathbf{S}) &= h(\mathbf{N}) - h(\mathbf{Z}|\mathbf{Z} + \mathbf{N}) \\
h(\mathbf{S}|\mathbf{Z}) - h(\mathbf{Z}|\mathbf{S}) &\geq h(\mathbf{N}) - h(\mathbf{Z}).
\end{aligned}
\tag{19}
$$

**Q.E.D**

If we link it back to the paper, $\mathbf{Z}$ is the novelty frontier reference points proposing distribution and $\mathbf{S}$ represents the achieved state distribution by executing goals sampled form $\mathbf{Z}$. $\mathbf{N}$ defines the behaviour of the policy according to a given reference point. When $\mathbf{N}$ has a Dirac delta distribution, the policy is a goal-conditioned policy (as in Skew-Fit). When $\mathbf{N}$ is a Gaussian distribution, the policy moves around the reference point following a Gaussian distribution. Compared to the goal conditioned policy, the reference point-conditioned policy achieves higher state entropy. The extra amount of entropy equals to the mutual information $I(\mathbf{Z} + \mathbf{N}; \mathbf{N})$.

# E  IMPLEMENTATION DETAILS

## E.1  ENVIRONMENT DETAILS

The three environments used in the experiments are implemented using Mujoco Todorov et al. (2012).

*PointMaze*: In this environment, an agent travels in a maze contains narrow passages. The observation is the $x$, $y$ position of the agent and the action is the velocity along the $x$, $y$ axis. The maximum velocity for each dimension is $0.12$. In the sparse reward experiment, the agent receives an extrinsic reward when the distance to the goal is less than $0.15$.

*DoorOpen*: In this environment, we use a single arm Yumi robot with 7 degrees of freedoms (DOFs). The robot is controlled in Cartesian space and the orientation of the end-effector is fixed. The robot can grab a door handle and open the door on a table. The maximum opening of the door is 90 degrees. The observation space is 5 dimensional, including the $x$, $y$, $z$ position of the end-effector, the open/close status of the gripper and the opening angle of the door. The action is the velocity along $x$, $y$, $z$ axis. The valid action space of the robot is $10cm \times 11cm \times 10cm$. In the sparse reward experiment, the agent receives an extrinsic reward when the angle difference to the goal is less than 2 degrees.

*OpenPressClose*: This environment contains a single arm Yumi robot, a box with a door and a button inside the box. The robot configuration is the same as the *DoorOpen* environment. The maximum opening of the door is 143 degrees. The observation space of the robot is 6 dimensional, including the $x$, $y$, $z$ position of the end-effector, the open/close status of the gripper, the opening angle of the door and the status of the button. The action is the velocity along $x$, $y$, $z$ axis. The valid action space of the robot is $2cm \times 28.5cm \times 16cm$. In the sparse reward experiment, the agent needs to press the button down for more than $1cm$ and close the door completely.

## E.2  HYPERPARAMETER

We use the same network structure for all experiments. The policy network contains 5 fully-connected layers with the number of units in all layers $32, 64, 128, 64, 32$. We use ReLU as the activation function and there is no activation for the output layer and for all experiments, the length of the trajectory is 200.

The RND contains 3 fully-connected layers with the number of units $32, 64, 64$ in the random target network and contains 4 fully-connected layers with the number of units $32, 64, 64, 128$ in the prediction network.

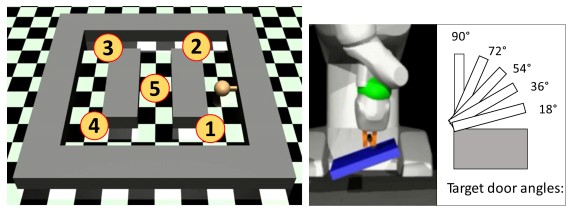

Figure 10: Five uniformly distributed points in PointMaze environment and the DoorOpen environment.

For PPO, we use the batch of $5,000$ and $15$ epochs per iteration. We update the state distribution and the goal distribution at every $5,000$ step.

## F    ADDITIONAL RESULTS

The second experiment test whether the algorithm is able to converge to a given sparse reward after exploration. As mentioned in the section 4.2, five points are selected to test the convergence of the algorithm given a sparse reward. Fig 10 shows the exact points in the PointMaze and DoorOpen environments. The points are selected uniformly from the area of interests. For each point, twelve experiments have been executed to receive information about mean and the variance of the algorithm's performance. Fig 11 and Fig 12 show the relationship between the convergence criteria and the number of iteration for PointMazz environment and DoorOpen environment respectively.

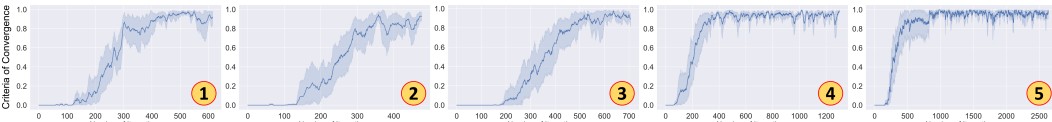

Figure 11: This figure show the relationship between convergence criteria and the number of iterations in PointMazz environment.

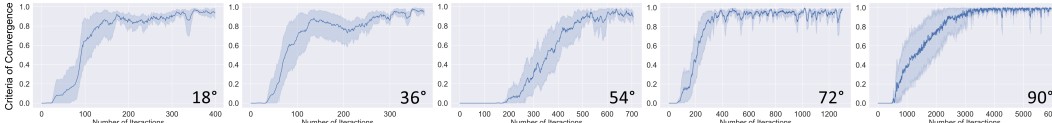

Figure 12: This figure shows the relationship between convergence criteria and the number of iterations in DoorOpen environment.

