# OpenReview forum: "Skew-Explore: Learn faster in continuous spaces with sparse rewards"
_ICLR.cc/2020/Conference — Reject_

### Official Review · AnonReviewer2 · 2019-10-22
**Official Blind Review #2**

**Rating:** 3

**Review:**

The authors propose an exploration objective for solving long-horizon tasks with sparse rewards. While the paper is heavily influenced by the recent Skewfit work (Pong et al, 2019),  it aims to solve a different problem (that of exploration for sparse reward RL), since Skewfit is interested in exploration and goal sampling for self-supervised (i.e. no external rewards) RL. The central idea in this paper is to encourage exploration by maintaining a distribution over the "current frontier", i.e. the set of points in the state space that are closed to the "edge" of what is explored by the policy , sampling points around this frontier, and encouraging the policy to reach these sampled points. The paper compares to RND (a method that is representative of state-of-the-art in exploration bonuses, I believe) and also to Skewfit, and outperforms these methods in two non-standard environments: door opening and point-mass navigation.

While the I think the empirical contributions are solid, and authors provide code to reproduce the results, I found the paper a bit hard to follow and understand, and different subsections in the technical section (Section 3) did not seem to have a unified narrative running through them. I will wait to see if other reviewers agree or disagree with me on this front, and if they do agree, then I think the paper will need substantial edits to improve its clarity before it is ready for publication.

**Experience Assessment:**

I have read many papers in this area.

**Review Assessment: Checking Correctness Of Derivations And Theory:**

I assessed the sensibility of the derivations and theory.

**Review Assessment: Checking Correctness Of Experiments:**

I assessed the sensibility of the experiments.

**Review Assessment: Thoroughness In Paper Reading:**

I read the paper at least twice and used my best judgement in assessing the paper.

---

> ### Author Response · Authors · 2019-11-15
> **Re: Official Blind Review #2**
>
> Thanks for your comment on our paper. We will try to improve the quality of the clarity in the next version.

---

### Official Review · AnonReviewer1 · 2019-10-23
**Official Blind Review #1**

**Rating:** 3

**Review:**

This paper proposes a new exploration algorithm by proposing a new way of generating intrinsic rewards. Specifically, the authors propose to maintain a "novelty frontier" which consists of states that have low-likelihood under some likelihood model trained on their replay buffer. The authors propose to sample from the novelty frontier using a scheme similar to a prior method called Skew-Fit, but replace the VAE with a kernel-based density model. To construct an exploration reward, the authors estimate the KL divergence between the resulting policy state distribution and the desired `state distribution, where the desire state distribution is a Gaussian centered around a point sampled from the novelty frontier.

Overall, the paper tackles an important questions of exploration, and while the concept of a frontier set is not novel, the authors propose a concrete instantiation that has promising results on continuous state spaces. I'm skeptical that this exact algorithm would work on domains with complex state spaces (e.g. images), where adding Gaussian noise to your state won't produce reasonable nearby states. That said, the general idea of fitting a new model to the latest trajectory and using KL as reward seems like a promising principle that could on its own scale. However, the theory seems a bit off and there are a few experimental details that make me hesitant to increase my score.

In details:

Theory:
I found the proof surprisingly long given that it amounts to saying that if (1) S = Z + N and (2) Z and N are independent, then
  H(S) >= H(N)
and so
  H(S | Z) - H(Z | S) = H(S) - H(Z) >= H(N) - H(Z)
Perhaps more worrisome is the statement, "we consider to maximize h(S|Z) - h(Z|S)". Unless I misread the paper, the authors do not maximize this quantity. Instead, they *fix* this quantity by choosing a fixed entropy of N. Worse yet, this quantity is actually minimizes since, while h(N) is fixed for the duration of the experiment, h(Z) is maximized ("To increase h(Z), we need to add..."). It would be good for the authors to address this concern, given that the claim of the paper is that they are "maximizing the entropy of the visited states." It seems like a simple answer is the following: given that S = Z + N, if N is fixed to some Gaussian distribution, then the authors simply need to maximize H(Z), which they are already doing. I'm not sure why the authors need to reason about H(S | Z) - H(Z | S).

Experiments:
Can Table 1 be replaced with the learning curves? The numbers 90% success and standard deviation of 3% seem like arbitrary numbers. It doesn't preclude the possibility that (e.g.) Skew-Fit or RND receives a 99% success rate with standard deviation of 3.1%. Figure 11 and 12 of the Appendix don't convince me that threshold at 90% and 3% is a particularly good choice.

Can the authors summarize the difference between coverage and entropy in the main paper? It seems like an important distinction. Given that the authors did not use all 8 pages, it would be good to explain it there rather than in the Appendix.
How sensitive is the method to the hyperparameter alpha? How was it chosen? Is it the same alpha chosen for Skew-Fit?
How was N chosen for the door environment?
Is Figure 7 (left) showing the performance on the simulated or real-world robot?
If it was done on the real-world robot, were there any important details in getting sim-to-real-world to work?
In Figure 5, why does there seem to be discrete jumps in the learning curves for "DoorOpen Coverage"?

I would be inclined to raise my score if:
1. The authors clarify why studying the quantity H(S | Z) - H(Z | S) is particularly important.
2. Address the concerns raised over the experiments.
3. Discuss more explicitly under what assumption they expect for this method (with a Gaussian KDE) to work well


**Experience Assessment:**

I have published in this field for several years.

**Review Assessment: Checking Correctness Of Derivations And Theory:**

I carefully checked the derivations and theory.

**Review Assessment: Checking Correctness Of Experiments:**

I carefully checked the experiments.

**Review Assessment: Thoroughness In Paper Reading:**

I read the paper thoroughly.

---

> ### Author Response · Authors · 2019-11-15
> **Re: Official Blind Review #1. Part 1**
>
> Thank you for the detailed review and comments. Below, we address your questions.
>
> --- 1. Why studying the quantity H(S | Z) - H(Z | S) is particularly important.
>
> The main difference between our work Skew-Explore and the prior work Skew-Fit is the behavior of the policy to be learned. In Skew-Fit, the policy is a goal-conditioned policy, meaning that, p(s|z) is a Dirac delta function centered at z. The entropy of the visited states equals the entropy of the proposed goals: h(S) = h(Z). In our work, we model the behavior of the policy given a sampled state p(s|z) as a Gaussian distribution centered at z. The entropy of the visited states is then: h(S) = h(Z) + h(S|Z) - h(Z|S).
> Comparing to the prior work, for the same goal proposing distribution, we gain an extra amount of entropy, which equals h(S|Z) - h(Z|S) = h(S) - h(Z) = I(S;Z) >=0. It means that, in every iteration, our method could visit more states, which leads to higher h(Z) and higher h(S) in the next iteration. Studying the quantity of h(S|Z) - h(Z|S) shows why our method could achieve better performance than the prior method.
>
> --- 2. Can Table 1 be replaced with the learning curves
>
> Thanks for your comment. We will try to improve the quality of the paper with a learning curve for representation of this experiment.
>
> --- 3. Can the authors summarize the difference between coverage and entropy in the main paper?
>
> Thanks for your comment on this point. There is not enough space in the main paper to clearly describe this point. We added a few sentences now in the appendix to explain their difference and how we computed them. In the next version, we will try to refine the paper by finding a proper place in the main paper to discuss this issue.
>
> --- 4. How sensitive is the method to the hyperparameter alpha? How was it chosen? Is it the same alpha chosen for Skew-Fit?
>
> We use alpha value as -1.1 for both Skew-Fit and our Skew-Explore experiments. Alpha equals -1 means that the skewed goal distribution is close to a uniform distribution over the visited state support range. Since the objective of this work is to maximize exploration, we choose a more aggressive strategy (alpha less than -1) for goal posing.
>
> --- 5. How was N chosen for the door environment?
>
> In the door environment, we model N as a Gaussian distribution with a standard deviation of 1.5.
>
> --- 6. Is Figure 7 (left) showing the performance on the simulated or real-world robot?
>
> Figure. 7 shows the performance in the simulation.
>
> --- 7. If it was done on the real-world robot, were there any important details in getting sim-to-real-world to work?
>
> We selected the policy with the best performance in the simulation and replayed the learned trajectory on the real robot. In order to make the sim-to-real to work, we need an accurate simulator of the Yumi robot and need to make the real scene close to the simulated scene as much as we can.

---

> > ### Author Response · Authors · 2019-11-15
> > **Re: Official Blind Review #1. Part 2**
> >
> >
> > --- 8. In Figure 5, why does there seem to be discrete jumps in the learning curves for "DoorOpen Coverage"?
> >
> > The discrete jumps happen in the coverage curve of the door opening experiment using Skew-Fit. The coverage measures how many states the agent has discovered at different iteration. In the door opening environment, the state has five dimensions: x, y, z of the gripper, gripper opening distance, and the door opening angle. The agent could quickly explore the full range of the first four dimensions (in the curve, the coverage increases rapidly at the beginning of the training). Then the increment of the coverage will slow down until the agent learned how to grab the door handle and open the door, and started to explore the last dimension of the state space.
> > Due to the nature of the door environment that the door opening angle dimension needs to be explored through a "grab the door handle" motion, which serves as a narrow passage in the state space. In order to explore states with different door opening angles, the goal proposing module must propose goal states with door opening angle larger than 0. However, at the early stage of the training process, we may not have enough goals proposed in those areas.
> > In our setup, every iteration, we collect trajectories with 25 goal states. Even though we skewed the goal distribution aggressively, most of the proposed goals are still located in the areas that can be reached without touching the door handle, which has little contribution to the coverage. When a promising goal is proposed, the agent has the chance to explore areas that have not been discovered yet. However, due to the limitation of the goal-conditioned policy used in Skew-Fit, the agent can only explore areas that are very close to the proposed goal, which contributes a small increment to the coverage curve. Then, the learning curve will be flat again until the next promising goal state is proposed. The whole process is represented as the "discrete jumps" in the coverage curve.
> > We do not observe similar patterns in our method because we allow the agent to have a broader exploration around the proposed goal. Once the robot grasps the door handle and starts to open the door, it will not just stop at a given angle but will try to move back and forth to open the door at different angles. The agent could explore the entire range of the door opening angle dimension in a few trials.
> >
> > --- 9. Discuss more explicitly under what assumption they expect for this method (with a Gaussian KDE) to work well.
> >
> > There are two places in our method that need density estimation,1) estimating the density of visited states p(s) to update the novelty frontier, and 2) computing log p_z(s) - log p_\tao(s) as the intrinsic reward. When we estimate the visited state density, we could replace KDE to other density estimation methods (such as VAE in Skew-Fit or flow-based methods), which scale well to high dimensional inputs. When we compute the intrinsic reward, we do not need an accurate density estimation. The key point is to construct an intrinsic reward that encourages the agent to reach the goal state but not staying at the goal state.
> > We could apply KDE on the lower-dimensional latent space we obtained while learning p(s). In this case, the learned distribution N on the raw space may not be a Gaussian distribution, but some other distribution centered at z. However, as long as the learned N has higher variance than the distribution learned with goal-conditioned policy (Dirac delta function), we always gain a positive h(S|Z) - h(Z|S), which gives additional power for exploration.

---

### Official Review · AnonReviewer4 · 2019-10-28
**Official Blind Review #4**

**Rating:** 3

**Review:**

This paper studies the problem of exploration in reinforcement learning. The key idea is to learn a goal-conditioned agent and do exploration by selecting goals at the frontier of previously visited states.  This frontier is estimated using an extension of prior work (Pong 2019). The method is evaluated on two continuous control environments (2D navigation, manipulation), where it seems to outperform baselines.

Overall, I like that the proposed method integrates some notion of novelty with the language of mutual information and density estimation. While this idea has been explored in prior work (as noted in the related work section), the proposed idea seems like a useful contribution to the literature. The use of convolutions to obtain a lower bound on mutual information seems neat. The experimental results are quite strong.

My main concern with the paper is a lack of clarity. I currently have enough reservations and questions (listed below) about the experimental protocol that I am learning towards rejecting this paper. However, if the paper clarified the concerns below, I'd be willing to increase by review.

Questions / Concerns:
* "[Prior work on mutual-information cannot] guarantee that the entire state space can be covered" -- In theory, I think these prior methods should cover the entire state space. Take the DIAYN objective, I(s, z) = H[s] - H[s | z]. This objective is maximized when p(s) is uniform over the entire state space and p(s | z) is a Dirac.
* "It is time consuming to collect enough samples to estimate an accurate state entropy" -- Can you provide a citation/proof for this? Why should we expect the proposed method to require fewer samples?
* "...entropy itself does not provide efficient information to adjust the action at each step." -- Can you provide a citation / proof for this? Also, what does "efficient information" mean?
* It seems like, if S is a finite collection of states in a continuous space, then it has measure zero, so its entropy should be zero. Can you explain why this is not the case?
* If I'm not mistaken, in equation 2, if we take (say) alpha = -0.5, then w_i is proportional to sqrt(p(s_i)), so w_i is an increasing function in p(s_i), not a decreasing function.
* Can you discuss how you might scale a KDE to high dimensions?
* "If the distribution has a larger range, the entropy is larger as well." -- Technically, this is not correct. You can construct distributions with larger ranges but smaller entropies.
* I think that Equation 3 should be the KL divergence between state marginal distributions, not between trajectories. If it were the KL between trajectories, it would include actions and policy terms.
* How are w_int and w_ext chosen? It seems like the method depends critically on the balance between these hyperparameters. Is w_int decayed over time? If not, why does the policy stop exploring once it has found the goal?
* What policy is used at convergence? It seems like the policy is conditioned on Z_t, so how is the Z_t chosen for evaluation?
* Fig 5 -- How are entropy and coverage computed? What are the maximum possible values for both of these quantities? What precisely does the X axis correspond to?
* Table 1 -- How did the baseline algorithms perform on this task?
* Fig 7 -- How did the baseline algorithms perform on this task? If the reward were sparse, shouldn't the Y axis be in the interval [0, 1]?
* "using coverage only during training is not suitable" -- Can you provide a citation/proof for this?
* "As a consequence, the entropy of the distribution of these points is also maximized" -- I believe that a finite number of points in a discrete space have measure zero, so they have zero entropy, regardless of the position of the points.



Other comments
* "What is the difference between *S* (in bold) and S_t?
* I would recommend using some notation other than p(s) to denote the smoothed/convolved density.
* "history states" -- I was confused about what this meant until it was introduced two sections later.
* "assimilate the definition of curiosity in psychology" -- I think that others (e.g., Oudeyer 2007, Pathak 2017) have noted the similarities between curiosity in humans and RL agents.
* Check for backwards quotes in the related work section.
* "Self-Goal Proposing" -- Some more related works are [Florensa 2017, Savinov 2018]
* "space associated environment" -- I don't know what this means.
* "disc rewards" -- I'd recommend spelling out discounted
* "truncated Gaussian function with a narrow range" -- Can you explain precisely what this is?
* In equation 2, I think it'd be clearer to write p^(1+\alpha).
* For the experiment on the effect of variance, I'd recommend making a plot instead of just listing the values.
* In Section 4.3, it's unclear whether the physical robot was successful at solving the task.
* "We rewrite the equation…" -- This paragraph is repeated.
* Double check that \citet and \citep are used properly

--------------UPDATE AFTER AUTHOR RESPONSE------------------
Thanks for answering many of my questions. This was helpful for clarifying my understanding. However, since a large fraction of my concerns were not addressed, so I am inclined with stick with my original vote to reject the paper. Nonetheless, I should emphasize that I think this paper is on the right track and the empirical results seems strong. With a bit more work on writing, I think it would be a fantastic paper at the next conference.

**Experience Assessment:**

I have published one or two papers in this area.

**Review Assessment: Checking Correctness Of Derivations And Theory:**

I assessed the sensibility of the derivations and theory.

**Review Assessment: Checking Correctness Of Experiments:**

I assessed the sensibility of the experiments.

**Review Assessment: Thoroughness In Paper Reading:**

I read the paper thoroughly.

---

> ### Author Response · Authors · 2019-11-15
> **Re: Official Blind Review #4. Part 1**
>
> Thank you for the detailed review. According to your comments and suggestions, we have added references, fixed typos, and removed/revised sentences with confusion or lack of citation/proof. Below, we address your questions.
>
> --- It seems like, if S is a finite collection of states in a continuous space, then it has measure zero, so its entropy should be zero. Can you explain why this is not the case?
>
> In our formulation, S_t is a finite collection of sampled history states in a continuous space, and S is a random variable with distribution estimated using techniques like weighted KDE from S_t.
>
> --- If I'm not mistaken, in equation 2, if we take (say) alpha = -0.5, then w_i is proportional to sqrt(p(s_i)), so w_i is an increasing function in p(s_i), not a decreasing function.
>
> The weight w_i does not need to be a decreasing function of p(s_i).In practice, there is a trade-off between exploration efficiency and learning efficiency which is controlled by w_i through parameter alpha. Proposing states with lower density is important for exploration, however, lower density also indicates that the agent are less trained on those states and may not know how to explore around them. Depending on the difficulty of the task and the environment, we could adjust alpha to decide how much we would like to emphasize the exploration.
> In our experiments, we set alpha as -1.1.
>
> --- Can you discuss how you might scale a KDE to high dimensions?
>
> There are two places in our method that need density estimation,1) estimating the density of visited states p(s) to update the novelty frontier, and 2) computing log p_z(s) - log p_\tao(s) as the intrinsic reward. When we estimate the visited state density, we could replace KDE to other density estimation methods (such as VAE in Skew-Fit or flow-based methods), which scale well to high dimensional inputs. When we compute the intrinsic reward, we do not need an accurate density estimation. The key point is to construct an intrinsic reward that encourages the agent to reach the goal state but not staying at the goal state.
> We could apply KDE on the lower-dimensional latent space we obtained while learning p(s). In this case, the learned distribution N on the raw space may not be a Gaussian distribution, but some other distribution centered at z. However, as long as the learned N has higher variance than the distribution learned with goal-conditioned policy (Dirac delta function), we always gain a positive h(S|Z) - h(Z|S), which gives additional power for exploration.
>
> --- "If the distribution has a larger range, the entropy is larger as well." -- Technically, this is not correct. You can construct distributions with larger ranges but smaller entropies.
>
> This sentence is related to the previous sentence. “The entropy of a continuous uniform function $U(p,q)$ is $\ln(p-q)$, and if the distribution has a larger range, the entropy is larger as well.” As a consequence, the distribution we meant here is continuous uniform distribution.
>
> --- I think that Equation 3 should be the KL divergence between state marginal distributions, not between trajectories. If it were the KL between trajectories, it would include actions and policy terms.
>
> The equation 3 indicates the KL divergence between the distribution formed by an actual trajectory, and the desired distribution modeled by N. Since we choose N to be a Gaussian distribution centered at a given reference state, we penalize states which are too far from the reference state, or states that the agent has stayed for too long.
>
> --- How are w_int and w_ext chosen? It seems like the method depends critically on the balance between these hyperparameters. Is w_int decayed over time? If not, why does the policy stop exploring once it has found the goal?
>
> In our experiments, we manually selected w_int and  w_ext to adjust the average intrinsic return to be around 1 and the extrinsic return to be 10. The w_int and  w_ext are fixed.
> The proposed z only affects the intrinsic reward, however, the overall objective is to optimize the return of the combined reward. Since the extrinsic return is much larger than the intrinsic return, once the agent has found the state with large extrinsic reward, the policy will eventually ignore the given goal but always go to the state with extrinsic reward.
>
> --- What policy is used at convergence? It seems like the policy is conditioned on Z_t, so how is the Z_t chosen for evaluation?
>
> In the evaluation, we randomly sample a z from the  Z_t and pass it to the policy. Since the policy has converged to a solution that always go to the state with extrinsic reward, it does not matter which z we send to the policy.

---

> > ### Author Response · Authors · 2019-11-15
> > **Re: Official Blind Review #4. Part 2**
> >
> >
> > --- Fig 5  How are entropy and coverage computed? What are the maximum possible values for both of these quantities? What precisely does the X axis correspond to?
> >
> > We added the computation of entropy and coverage to Appendix B.
> > We estimate the entropy and the coverage via uniform discretization over the environment state spate. We do not check the validity of the discretized states, and there will be states that are not reachable from the initial state (such as the obstacle area in PointMaze environment). The maximum possible value for both entropy and coverage are unknown.
> > The X axis represents the training iteration. One iteration contains 5,000 steps.
> >
> > --- Table 1 -- How did the baseline algorithms perform on this task?
> >
> > The results of the baseline methods are not reported because they could not even discover all 5 target states.
> >
> > --- Fig 7 -- How did the baseline algorithms perform on this task? If the reward were sparse, shouldn't the Y axis be in the interval [0, 1]?
> >
> > The baseline methods also failed on this task. The Y axis is the extrinsic reward times w_ext, which is in the interval [0, 10].
> >
> > --- "As a consequence, the entropy of the distribution of these points is also maximized" -- I believe that a finite number of points in a discrete space have measure zero, so they have zero entropy, regardless of the position of the points.
> >
> > We treat the points as samples from a distribution, and measure the entropy of this distribution that we draw samples from.
> >
> > --- "What is the difference between *S* (in bold) and S_t?
> >
> > S_t is a finite collection of sampled history states in a continuous space, and S is a random variable with distribution estimated using techniques like weighted KDE from S_t. We can say that S_t contains samples from S.
> >
> > --- "truncated Gaussian function with a narrow range" -- Can you explain precisely what this is?
> >
> > In probability and statistics, the truncated normal distribution is the probability distribution derived from that of a normally distributed random variable by bounding the random variable from either below or above (or both)[1].  In our case, the normal is bounded as  \mu-\epsilon<x<\mu-\epsilon, where \ epsilon is an arbitrary small number. https://en.wikipedia.org/wiki/Truncated_normal_distribution
> >
> > --- In equation 2, I think it'd be clearer to write p^(1+\alpha).
> >
> > Thanks for your comment. In equation 2, we would like to show that the p^(\alpha) is acting as a weight on the p.

---

### Decision · Program_Chairs · 2019-12-19

**Decision:**

Reject

**Comment:**

While the reviewers generally appreciated the ideas presented in the paper and found the overall aims and motivation of the paper to be compelling, there were too many questions raised about the experiments and the soundness of the technical formulation to accept the paper at this time, and the reviewers did not feel that the authors had adequately addressed these issues in their responses. The main concerns were (1) with the correctness and rigor of the technical derivation, which the reviewers generally found to be somewhat questionable -- while the main idea seems reasonable, the details have a few too many question marks; (2) the experimental results have a number of shortcomings that make it difficult to fully understand whether the method really works, and how well.